# Persistent Thyroid Carcinoma and Pregnancy: Outcomes in an Italian Series and Review of the Literature

**DOI:** 10.3390/cancers14225515

**Published:** 2022-11-10

**Authors:** Carla Colombo, Simone De Leo, Noemi Giancola, Matteo Trevisan, Daniele Ceruti, Francesco Frattini, Luca Persani, Laura Fugazzola

**Affiliations:** 1Department of Endocrine and Metabolic Diseases, Istituto Auxologico Italiano IRCCS, 20149 Milan, Italy; 2Department of Pathophysiology and Transplantation, University of Milan, 20122 Milan, Italy; 3Department of Medical Biotechnology and Translational Medicine, University of Milan, 20100 Milan, Italy; 4Division of Surgery, Istituto Auxologico Italiano IRCCS, 20095 Milan, Italy

**Keywords:** pregnancy, thyroid carcinoma, thyroglobulin, outcome, persistence, DTC, delivery

## Abstract

**Simple Summary:**

Since scanty data are available on the progression risk in patients with persistent thyroid cancer (TC) who undergo pregnancy, we investigated this topic in our series of patients followed up in a tertiary care thyroid cancer center and performed a review of the literature. We found that pregnancy is not associated with disease progression in patients with stable local and/or distant persistence before conception. A transient increase in thyroglobulin levels can be observed during pregnancy, but they return to pre-conceptional levels after delivery. Thus, pregnancy should not be contraindicated even in patients with distant metastases, though a precise clinical characterization, including the disease stage at diagnosis, the ATA risk class, and the dynamic risk stratification, is recommended.

**Abstract:**

Scanty data are available on the progression risk in patients with persistent thyroid cancer (TC) before pregnancy. We aimed to evaluate this topic in our series and to review available literature data. This was a retrospective study performed in a tertiary care Italian TC center. We included 8 patients with persistent papillary TC who became pregnant after initial treatments (mean time interval of 62 months). Seven patients had the structural disease (lung and/or neck node metastases), while one patient had biochemical persistence. During a mean follow-up of 97 months, none of the patients showed disease progression either during pregnancy or during a follow-up of at least 12 months after delivery, and no additional treatments were needed. A sequential biochemical evaluation showed that thyroglobulin levels can significantly increase during pregnancy, returning to preconception levels after delivery. In conclusion, our data confirm that pregnancy is not associated with disease progression in patients with stable local and/or distant persistence before conception. Thus, pregnancy should not be contraindicated in metastatic women, although a precise clinical characterization, including the disease stage at diagnosis, the ATA risk class, and the dynamic risk stratification, should be conducted before conception.

## 1. Introduction

Differentiated thyroid cancer (DTC) is the second most frequently diagnosed during pregnancy and the postpartum period, with an incidence of approximately 14 cases per 100,000. In about 75% of cases, diagnosis is conducted in the post-partum, while in 25%, it is performed in the prenatal period [1,2]. It is well known that, during pregnancy, the overall thyroid volume and the size of benign and malignant thyroid nodules increase [3,4], likely due to the effect of high growth factor levels. On the other hand, the potential role of pregnancy on the final outcome in women with cured or persistent TC is highly debated [5]. As far as patients in remission concerns, available studies show that the disease outcome does not significantly change during pregnancy and the postpartum period, indicating that these women should only be followed by routine assessment of thyroglobulin (Tg) and anti-thyroglobulin autoantibodies (TgAb) levels [6]. By contrast, for women with biochemical or structural persistence, the available studies are more limited, and the data are controversial. To date, only five studies have been reported on this topic, and the results indicate that pregnancy can occasionally be associated with a slight increase in already existing metastatic lesions, though without a significant impact on progression and mortality risk [7,8,9,10,11]. The study, including the largest series of persistent patients, showed that none of the patients with an excellent, indeterminate, or biochemical incomplete response to therapy prior to pregnancy developed structural persistence during pregnancy or postpartum, while women with structural disease persistence before pregnancy were more likely to have structural disease progression (16%), 8% of them requiring additional therapy [9]. Unfortunately, the majority of these cohorts are limited in size, the definition of persistence is not adequately described in most cases, as well as the precise biochemical and imaging trend during pregnancy. Thus, to date, there is no clear scientific consensus for the management of patients with persistent DTC during pregnancy, limiting the possibility of reliable counseling in persistent women willing to be pregnant.

Aiming to obtain more insights into this topic, we analyzed a group of patients with persistent DTC before pregnancy followed in our tertiary center and performed a comprehensive review of all the data available in the literature. The original data of our study come from the tight biochemical and ultrasonographic evaluation of the patients included, which allowed us to follow the tumor behavior during pregnancy and in the years after delivery.

## 2. Materials and Methods

### 2.1. Patients

Among 1200 patients followed by a single tertiary care endocrine center during the period of 1990–2022, we retrospectively selected patients who had a pregnancy after initial treatment in the period of 2003–2021. Only patients with DTC and biochemical and/or structural persistence, according to Tuttle et al. [12], within the 12 months before pregnancy and a follow-up after delivery of at least 12 months were included. Exclusion criteria were poorly differentiated or anaplastic TCs, patients with pregnancy and delivery that occurred before or during the diagnosis of TC, patients with a diagnosis of TC before pregnancy but with confirmed disease remission at the study time, and patients with a follow-up shorter than 12 months.

All included patients were treated with total thyroidectomy (TT) associated or not with lymph-node neck dissection and radioiodine residue ablation (RRA). Each patient was risk-stratified according to the 8th edition of the American Joint Commission on Cancer (AJCC) [13] and to the American Thyroid Association (ATA) 2015 Guidelines [14]. The response to initial therapy (excellent, indeterminate, biochemical incomplete, or structural incomplete) during the first 24 months of follow-up was based on the dynamic risk classification (DRS) [12].

Levothyroxine dose was titrated in order to maintain TSH at the levels suggested for the ATA risk class and according to the DRS. Patients were followed up 1 month after surgery, thereafter every 6 months by a full evaluation including physical examination, thyroglobulin (Tg) and anti-thyroglobulin autoantibodies (TgAb) measurement, and neck ultrasound always performed by the same operator (LF). In particular, Recombinant human thyrotropin (rhTSH)-stimulated Tg was evaluated up to the first months of 2016 when we started to assess patients only by basal ultrasensitive Tg (Elecys Tg II-Roche Diagnostics, Basilea-Switzerland, analytical sensitivity 0.04 mcg/L). For anti-thyroglobulin autoantibodies, Elecsys^®^ anti-Tg Roche Diagnostics (upper limit normal 115 kU/L) was used.

In order to identify remission or persistent/recurrent disease, ATA guidelines and Italian consensus recommendations were followed [14,15]. In particular, lymph-nodal metastases were diagnosed by neck ultrasound and by means of the thyroglobulin measurement in the wash out- fluid at fine needle aspiration. Distant metastases were diagnosed by radioiodine uptake or, in the case of radioiodine refractoriness, by computed tomography (CT)/magnetic resonance imaging (MRI)/fluorodeoxyglucose-positron emission tomography (FDG PET) imaging.

The Institutional review board approved the study (#THY-CANC, 2022_03_08_03), and informed patient consent for the collection of clinico-pathological data during follow-up was always obtained.

### 2.2. Molecular Characterization

The tumor tissue was available for patients #5, #6, and #7; DNA and total RNA were extracted by standard methods, and DNA and cDNA were analyzed using the custom PTC-MA assay based on matrix-assisted laser desorption/ionization time-of-flight mass spectrometry, and previously set up for the simultaneous identification of 19 known genetic alterations [16].

### 2.3. Review of the Literature

We performed a PubMed search, including the terms pregnancy, thyroid carcinoma, and persistence. Moreover, the references in each included paper and in all available Guidelines were also checked to identify additional relevant studies.


*Inclusion Criteria for Studies*


-Studies that evaluated the potential pregnancy effect on disease outcomes in patients with DTC (i.e., papillary and follicular) in persistence before pregnancy;-Studies in which the authors followed up with patients according to ATA international guidelines.


*Exclusion Criteria for Studies*


-Opinions, reviews, commentary, case reports, and insufficient data;-Studies including thyroid only tumors other than DTCs;-Studies including patients with a diagnosis of TC following pregnancy and/or delivery;-Studies including only patients affected with TC in disease remission;-Studies written in languages other than English.

## 3. Results

### 3.1. Clinicopathological and Genetic Characteristics of Patients Included

We report data of eight consecutive patients with PTC (7 conventional variant PTCs and 1 sclerosing variant PTC) with a mean age at diagnosis of 27.6 ± 5.5 years (yrs) (range of 21–35) and a mean time between PTC diagnosis and pregnancy of 62 ± 43.8 months (range of 12–120) (Table 1).

Among the seven patients with the structural disease, two patients had lung metastases (bilateral micronodular dissemination in both cases), diagnosed by FDG PET-CT in patient #1 and by whole-body scan and CT scan in patient #3. In five patients, neck lymph node recurrences were diagnosed and confirmed at the time of pregnancy onset; only one patient showed biochemical persistence without radiological evidence of disease (Table 2).

In accordance with the 8th edition of the AJCC and the 2015 ATA Guidelines, 75% of women included had an AJCC Stage I and an intermediate risk of recurrence, whereas 25% of them had an AJCC Stage II and a high ATA risk of recurrence (Table 1 and Table 2). The evaluation of DRS at 24 months of follow-up showed a structural incomplete response in 7 patients and a biochemical incomplete response in 1 woman (Table 1).

All eight patients analyzed had a stable, persistent disease before conception. Anti-Tg autoantibodies were negative in all cases.

Patients with distant metastases (#1 and #3) were maintained on TSH suppressive treatment during the gestational period, while those with local recurrences or biochemical persistence were on TSH semi-suppressive therapy (Table 2). Among the three cases with a genetic characterization, #5 had BRAFV600E mutation, #7 harbored a double BRAFV600E and TERTc-124C > T mutation, and #6 was wild type for the alterations analyzed.

It is worth noting that all the included patients had a normal gonadal function and that, in all cases, the delivery was eutocic without fetal complications.

### 3.2. Evaluation of Structural and Biochemical Disease Progression during and after Pregnancy

During a mean follow-up of 97 months (range of 12–212), none of the patients showed structural disease progression either during pregnancy or during a follow-up of at least 12 months after delivery, and no additional treatments were needed.

In detail, patient #1, with a diagnosis of PTC in Graves’ diseases and radioiodine refractory lung metastases, showed during pregnancy variable but stable Tg and TSH receptor auto-antibodies (TRAb) levels (Figure 1).

On the contrary, the second patient with lung metastases (#3) showed an increase in Tg compared to pre-pregnancy values (119 vs. 41 ng/mL) but with a progressive reduction during post-partum follow-up with values ranging from 40 to 60 µg/L (Figure 1). In case #1, TRAb levels reduced during pregnancy, while in case of #3, Tg levels increased in association with a TSH increase in the very first phases after conception but remained elevated in the following period despite TSH-suppressed levels. In both patients, radiological stability was demonstrated by a PET CT scan (#1) and CT scan (#3) performed in the postpartum period.

Four out of five patients with loco-regional lymph nodal persistence showed stable Tg levels during pregnancy, and no variations in lymph node volume were detected at the ultrasound evaluations performed periodically during pregnancy and after delivery (exemplificative patient #2 in Figure 2).

The remaining case (#4) showed a slight Tg increase in the post-partum period (2 vs. 0.6 µg/L), which paralleled TSH variations, with the stability of cervical lymph node metastases size (Figure 2). Finally, the patient with biochemical persistence of disease (#7) showed stable Tg levels during pregnancy and after delivery (mean Tg values of 0.4 µg/L).

### 3.3. Review of the Literature

We reviewed the five studies published to date evaluating the potential effect of pregnancy on disease outcomes in patients with persistent DTCs after initial treatments (Table 3).

Some series also included not-pregnant women, who have been excluded from our review to make data exclusively related to the present topic. In particular, 2 series are from the USA (number of patients 36 and 235), 1 from Israel (13 persistent patients), 1 from Morocco (12 persistent patients), and 1 from Japan (28 patients). No studies including European series have been published to date.

The complete TNM assessment was performed in three out of five post-surgery studies [8,11], and only three out of five studies assessed ATA risk for patients [9,10], including the present one. The DRS, which evaluates the response to therapy after initial treatments, was evaluated only in the large American series [9] and in the present study. Among the available studies, biochemical or structural progression after pregnancy ranged from 0 to 15% and from 8 to 31%, respectively. The mean follow-up was not reported in 2 studies [8,9] and was of few months in 2 additional studies [7,10], while long follow-ups are available only for the Yamazaki’s series (131 months) [11] and for the present study (97 ± 79.4).

## 4. Discussion

In the present series of patients with persistent local and distant disease at the time of conception, no biochemical or structural progression was found either during pregnancy or at a follow-up of at least 12 months after delivery. By sequentially measuring the tumor marker, we observed that in two patients, Tg levels significantly increased during pregnancy, at returned to the preconception levels after delivery. This finding indicates that the biochemical progression does not represent a natural course of the disease, but it is directly related to the growth stimulus of pregnancy. In this context, we demonstrated in a large series of female and male patients with PTC that estrogen receptors α (ERα) and progesterone receptor (PR) were expressed in more than 65% of tumor tissues and correlated with larger tumor size at diagnosis [17]. In some patients, we observed an increase in TSH levels in the first period after conception, and a consensual Tg increase was found. Nevertheless, the elevation of Tg levels was maintained even after TSH suppression by L-thyroxine titration, confirming that pregnancy-related growth factors actually stimulate Tg secretion. In this context, it is worth noting that the lack of progression was observed in patients on TSH, either suppressive or semi-suppressive L-thyroxine treatment, and no pregnancy, delivery, or fetal clinical complications occurred. This indicates that even the TSH suppressive treatment could be maintained, especially in the presence of distant metastases, though a tight thyroid function evaluation is recommended.

The results of our study agree with the few available in the literature, all coming from non-European Centers [7,8,9,10,11], and here reviewed, confirming that the vast majority of patients with known biochemical or structural persistence of disease at the time of conception remain stable or slowly progresses during pregnancy.

Interestingly, the lack of progression was documented even in one patient with lung metastases, and TRAb titers were still high nine years after initial treatment. Those Ab were thus elicited towards the lung metastases, and we were afraid that they may contribute to tumor growth during pregnancy. Fortunately, likely due to the physiological decrease in autoimmunity in the gestational period, the clinical picture remained stable, indicating that these peculiar patients are not at higher risk of progression during pregnancy.

The major drawback of this study is the low number of cases included, though patients were fully characterized with biochemical parameters precisely documented and the disease response to treatment assessed. Moreover, this is the only available European study with follow-up > 5 years, this being a crucial aspect when considering tumors with a low growth rate like. Nevertheless, our results cannot be generalized to all patients with persistent disease undergoing pregnancy and need to be further confirmed by data obtained in other, possibly larger, series.

## 5. Conclusions

In conclusion, our data, obtained in a small and well-characterized series, the first one including European patients, support previous findings and confirm that even patients with distant metastases can safely undergo pregnancy. Interestingly, we demonstrated for the first time that the biochemical progression observed during gestation can be transient, thus implying a direct effect of pregnancy. Metastatic women willing to be pregnant should have a precise clinical characterization, including the disease stage at diagnosis, the ATA risk class, and the dynamic risk stratification, and should be informed that a minor risk of biochemical or structural disease progression exists. This recommendation should probably be limited to the more differentiated forms of TC, whereas additional studies are needed to document the behavior of aggressive tumors, such as poorly differentiated TC or advanced medullary TC, upon the exposure of pregnancy growth factors.

## Figures and Tables

**Figure 1 cancers-14-05515-f001:**
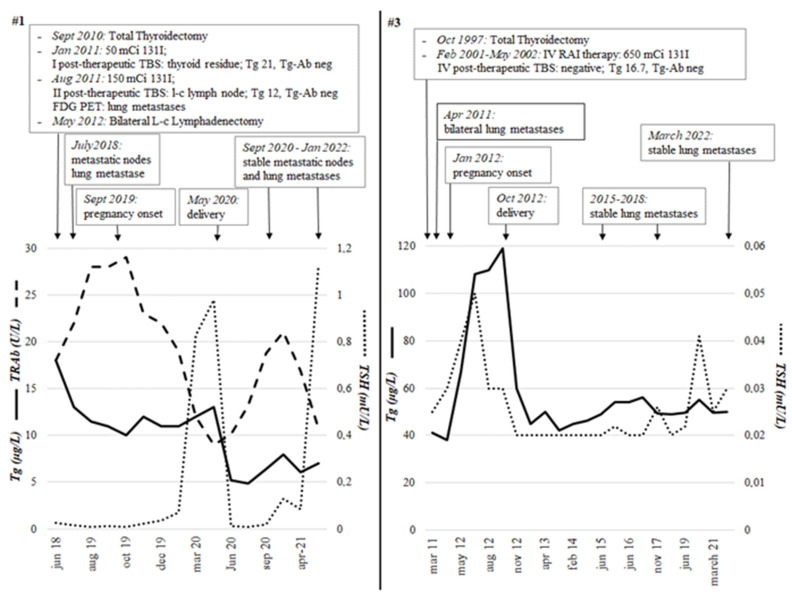
The two patients (#1 and #3) with papillary thyroid carcinoma and distant metastases at the time of pregnancy onset. The trend of Tg, TSH, and TRAb from the initial treatment (thyroidectomy and radiometabolic therapy) to the last follow-up visit, including pregnancy and postpartum period, is shown. Both patients had negative anti-thyroglobulin auto-antibodies. Both patients had no structural disease progression during or after pregnancy. Tg, Thyroglobulin; TSH, thyroid-stimulating hormone; TRAb, TSH receptor auto-antibodies; TBS, Total Body Scintigraphy; mCi, millicurie; RAI, Radioactive iodine.

**Figure 2 cancers-14-05515-f002:**
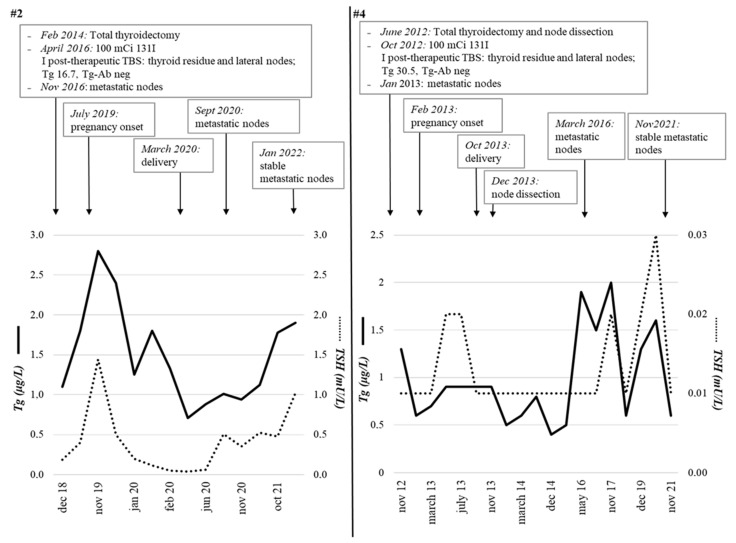
Two representative patients (#2 and #4) with papillary thyroid carcinoma in persistent disease (neck node metastases) at the time of pregnancy onset. The trend of Tg and TSH from the initial treatment (thyroidectomy and radiometabolic therapy) to the last follow-up visit, including pregnancy and postpartum period, is shown. Both patients had negative anti-thyroglobulin auto-antibodies. Both patients had no structural disease progression during or after pregnancy. Tg, Thyroglobulin; TSH, thyroid-stimulating hormone; TBS, Total Body Scintigraphy; mCi, millicurie; RAI, Radioactive iodine.

**Table 1 cancers-14-05515-t001:** Risk classifications and age data at different time points assessed in PTCs series.

Age at PTC Diagnosis (yrs)		Age at Eutocic Delivery (yrs)		Time between PTC Diagnosis and Pregnancy (yrs)		ATA 2015 Risk Stratification *		Dynamic Risk Stratification **	
Mean ± SD	27.6 ± 5.5	Mean ± SD	33.1 ± 4.6	Mean ± SD	62 ± 43.8	Low Risk	0/8	Biochemical incomplete	1/8
Median	27	Median	35	Median	66	Intermediate Risk	6/8	Structural incomplete	7/8
Range	21–35	Range	27–38	Range	12–120	High Risk	2/8		

Legend: PTC, Papillary Thyroid Carcinoma; SD, Standard Deviation; ATA, American Thyroid Association; years, yrs; * according to [14], ** according to [12].

**Table 2 cancers-14-05515-t002:** Clinicopathological features of 8 PTC patients with Structural or Biochemical Disease Persistence before pregnancy.

Patient ID	Age at D (yrs)	Histology, Tumor Size (mm)	TNM, 8th AJCC Stage	Surgery	RAI (Total I131 dose-mCi)	LT4 Therapy (TSH Target)	Final Disease Outcome	Metastatic Site	Disease Progression after Pregnancy	FU from TT (Months)	FU from Eutocic Delivery (Months)
#1	27	CPTC, 8	pT1bNXM1 (II)	TT	200	<0.1 mU/L	sD	DM (lung)	No	194	134
#2	29	SCPTC, 5	pT1aNXM0 (I)	TT	100	0.1–0.5 mU/L	sD	LM (neck)	No	272	212
#3	22	CPTC, 13	pT3N1M1 (II)	TT + L	650	<0.1 mU/L	sD	DM (lung)	No	69	12
#4	35	CPTC, 28	pT2N1bM0 (I)	TT + L	100	0.1–0.5 mU/L	sD	LM (neck)	No	90	78
#5	24	CPTC, 9	pT1aNXM0 (I)	TT	150	0.1–0.5 mU/L	sD	LM (neck)	No	186	174
#6	36	CPTC, 22	pT2N1bM0 (I)	TT + L	230	0.1–0.5 mU/L	sD	LM (neck)	No	113	12
#7	27	CPTC, 26	pT2N1bM0 (I)	TT + L	170	0.1–0.5 mU/L	bD	-	No	237	141
#8	21	CPTC, 23	pT2N1aM0 (I)	TT + L	200	0.1–0.5 mU/L	sD	LM (neck)	No	64	14

Legend: D, Diagnosis; yrs, years; PTC, Papillary Thyroid Carcinoma; CPTC, Conventional PTC; SCPTC, Sclerosing variant PTC; AJCC, American Joint Committee on Cancer; TT, Total Thyroidectomy; L, Lymphadenectomy; RAI 131I, treatment with radioiodine; mCi, millicurie; LT4, levothyroxine; bD, biochemical persistence of Disease; sD, structural persistence of Disease; LM, Lymph node Metastases; DM, Distant Metastases; FU, Follow Up.

**Table 3 cancers-14-05515-t003:** Revision of clinical studies published to date on DTCs patients in disease persistence before pregnancy.

References	Leboeuf et al., 2007 [7]	Hirsch et al., 2010 [8]	Rakhlin et al., 2017 [9]	Driouich et al., 2021 [10]	Yamazaki et al., 2022 [11]	Present Study
**Country**	USA	Israel	USA	Morocco	Japan	Italy
**Study design**	Retrospective	Retrospective	Retrospective	Retrospective	Retrospective	Retrospective
**Number of patients** *(patients in persistence/total of patients)*	NA/36	13/23	87/235	12/42	28/28	8/8
**Total of pregnancies**	36	90	235	42	55	10
**Age at diagnosis of TC** *(yrs)* mean ± SD *(range)*	28.1 ± 4.6 (19.9–37.4)	24.7 ± 5.8 (10–38)	28 ± 7 (16–43)	/	25 (4–41)	27.6 ± 5.5 (21–35)
**Age at delivery** *(yrs)* mean ± SD *(range)*	33.6 ± 3.9 (24.3–42.2)	29.7 ± 4.7 (20.1–41.2)	34 ± 0.4 (20–45)	/	32 (25–45)	33.1 ± 4.6 (27–38)
**Histotype** *(PTC/FTC/other)*	34/1/1	63/0/0	219/6/10	42/0/0	23/5/0	8/0/0
**Tumor size** *(mm)* mean ± SD *(range)*	23 ± 14 (7–68)	/	21 ± 1 (2–85)	/	/	16.7 ± 9 (5–28)
**pTNM**						
**pT** *T1/T2/T3/T4/NA*	/	30/11/17/1/4	/	22/10/6/4/0	11 (T1-T2)/11 (T3-T4)/6 (NA)	3/4/1/0/0
**pN** *N0-NX/N1/NA*	13/17/6	30/31/2	114/115/6	24/10/8	8/20/0	3/5/0
**pM** *M0/M1/NA*	31/1/4	59/2/2	225/10/0	39/3/0	13/15/0	6/2/0
**AJCC stage** *(I/II)*	/	/	225/10	/	/	6/2
**ATA** Risk *(Low/Int/High)*	/	/	37/184/14	22/14/6	/	0/6/2
**First Surgery** *(TT/TL)*	29/7	59/4	210/25	/	28/0	8/0
**RAI therapy** *(n.)*	23	58	144	9	28	8
**Total I-131 dose** *(mCi, median)*	/	150	/	/	200	185
**Additional treatments** *(LND/TKI)*	1/0	/	2/1	/	0/1	1/0
**Response to therapy (DRS)** *Excellent/Ind/BIR/SIR*	/	/	148/29/20/38	/	/	0/0/1/7
**Time between TC diagnosis and pregnancy** *(yrs)* *mean ± SD*	5.5 ± 3.8	5.08 ± 4.39	4.9 ± 0.3	4.4 ± 3.1	/	5.16 ± 3.65
**Follow-up from pregnancy** *(months)* *median/mean ± SD (range)*	4 (1.2–20.2)	/	/	14.45 ± 2.4	131 (50–390)	97 ± 79.4 (12–212)
**sTg level before pregnancy** (*ng/mL)*						
*mean* *± SD/median (range)*	1.2	1.23 ± 3.93 (0–23.9)	3.8 ± 27	1.75 (0–12.4)	0.029 (0.001–25.06)	1.7 (0.2–39)
**sTg level after pregnancy** *(ng/mL)*						
*mean* *± SD/median (range)*	1	1.18 ± 3.54 (0–21.4)	3.7 ± 15	1.98 (0–18.5)	/	1.4 (0.2–37)
**Biochemical/structural persistence** *(disease site)*	NA/5 (4 cervical LFN MTS + 1 lung MTS)	6/7 (7 cervical LFN MTS)	20/38 (7 lung + 31 cervical LFN MTS)	NA/12 (9 cervical LFN MTS + 3 distant MTS)	0/28 (26 lung + 2 lung and bone MTS)	1/7 (5 cervical LFN MTS + 2 lung MTS)
**Stable disease/** **biochemical/structural progression after pregnancy** *(in patients with disease persistence)*	4/NA/1 (80/NA/20%)	7/2/4 (54/15/31%)	27/NA/11 (71/NA/29%)	11/0/1 (92/0/8%)	25/0/3 (89/0/11%)	8/0/0 (100/0/0%)

Legend: TC, Thyroid Cancer; PTC, Papillary Thyroid Carcinoma; FTC, Follicular Thyroid Carcinoma; yrs, years; NA, Not Available; RAI: radioactive iodine; mCi, millicurie; yrs, years; AJCC, American Joint Committee on Cancer; ATA, American Thyroid Association; Int, Intermediate; TT, Total Thyroidectomy; LT, Lobectomy; LND; Lymphadenectomy; TKI, Tyrosine kinase Inhibitor; DRS, Dynamic Risk Stratification; BIR, Biochemical Incomplete Response; SIR, Structural Incomplete Response; sTg; serum Thyroglobulin (excluding patients with positive anti-thyroid autoantibodies); LFN MTS, Lymph node metastases.

## Data Availability

The authors confirm that the data supporting the findings of this study are available within the article.

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
