# Peer review of "Persistent Thyroid Carcinoma and Pregnancy: Outcomes in an Italian Series and Review of the Literature"

_cancers, 2022, doi:10.3390/cancers14225515_

Round 1
Reviewer 1 Report
I’m glad to review this manuscript titled “Persistent thyroid carcinoma and pregnancy: outcomes in an Italian series and review of the literature.” In this case series and review of literature, authors tried to clarify whether the development and/or recurrence risk of TC associated pregnancy.
This study included 8 women after delivery, and all of them had a pregnancy after initial TC treatment. They have biochemical and/or structural persistence. However, during a mean follow-up of 97 months, none of the patients showed structural disease progression either during pregnancy or during a follow-up of at least 12 months after delivery, and no additional treatments were needed.
This is a valuable because authors described the clinical courses of each case in detail; therefore, this series contribute for clinicians who routinely treat pregnant women with thyroid cancer.
However, unfortunately, it is too small study population to clarify the general outcomes of young women with TC associated pregnancy as well as after delivery. As authors know, the time to recurrence is very long in young thyroid cancer patients.
There are some major criticisms which should be addressed.
Major comments 1.
It is unclear whether pregnancy is associated with disease progression in patients with stable local and/or distant persistence before conception. Number of cases is too small to conclude it.
Please discuss whether this series reflects the general course of thyroid carcinoma complicated by pregnancy.
Major comments 2.
Authors conclude that strict follow up is recommended. This statement may not reflect the authors' view that pregnancy does not influence disease progression. Please describe why strict follow up is required.
Major comments 3.
Among the 7 patients, 2 patients had lung metastasis and 5 had lymph node metastasis. Please describe how to diagnosis these metastases. Do authors have histological evidences?
Major comments 4.
In Figure 1, why Tg of case #3 have been increased during pregnancy.
Major comments 5.
As the authors know, Tg is not accurate in the presence of TgAb. Are all cases in this series negative for TgAb?
Author Response
Comments to the Author
I’m glad to review this manuscript titled “Persistent thyroid carcinoma and pregnancy: outcomes in an Italian series and review of the literature.” In this case series and review of literature, authors tried to clarify whether the development and/or recurrence risk of TC associated pregnancy.
This study included 8 women after delivery, and all of them had a pregnancy after initial TC treatment. They have biochemical and/or structural persistence. However, during a mean follow-up of 97 months, none of the patients showed structural disease progression either during pregnancy or during a follow-up of at least 12 months after delivery, and no additional treatments were needed.
This is a valuable because authors described the clinical courses of each case in detail; therefore, this series contribute for clinicians who routinely treat pregnant women with thyroid cancer.
However, unfortunately, it is too small study population to clarify the general outcomes of young women with TC associated pregnancy as well as after delivery. As authors know, the time to recurrence is very long in young thyroid cancer patients. There are some major criticisms which should be addressed.
Major comments 1: It is unclear whether pregnancy is associated with disease progression in patients with stable local and/or distant persistence before conception. Number of cases is too small to conclude it. Please discuss whether this series reflects the general course of thyroid carcinoma complicated by pregnancy.
R1: All patients had indeed stable disease persistence before conception and did not show progression during pregnancy and in the postpartum period. Nevertheless, although carefully studied and possibly adding new data to the literature, the cohort is small and the results obtained cannot necessarily be generalized to other cohorts of patients with thyroid cancer during pregnancy, as observed by the Reviewer. This point has been better highlighted in the Discussion (page 9, lines 269-271).
Major comments 2: Authors conclude that strict follow up is recommended. This statement may not reflect the authors' view that pregnancy does not influence disease progression. Please describe why strict follow up is required.
R2: We agree with the Reviewer's comment. Indeed, according to our results, which confirm data available in the literature, these patients can have the same follow up of other persistent not-pregnant patients (with the exception of the monthly TSH measurement for L-thyroxine titration). Nevertheless, we believe that a precise clinical characterization of the metastatic disease is crucial in these cases before conception. We changed the text accordingly (see last sentences of the simple summary and abstract).
Major comments 3: Among the 7 patients, 2 patients had lung metastasis and 5 had lymph node metastasis. Please describe how to diagnosis these metastases. Do authors have histological evidences?
R3: As described in Methods, we assessed disease persistence in accordance with ATA guidelines and Italian consensus recommendations [14, 15] (see page 3, lines 101, 102).
In particular, neck lymph node metastases were diagnosed by Tg measurement in the wash-out fluid, while lung metastases were diagnosed at the WBS performed during radiometabolic treatment, then followed at chest CT scans or by FDG-PET CT scan. This information has been added to the Methods and Results sections (see page 3, lines 102-107 and page 4, lines 144-147).
Major comments 4: In Figure 1, why Tg of case #3 have been increased during pregnancy.
R4: The Tg pattern in case #3 has been described in the text (page 6, lines 187-192). This patient had a Tg increase paralleling an increase in TSH levels in the very first phases after conception, and remained elevated in the following period despite TSH suppressed levels. After delivery Tg returned to preconceptional levels. We believe that the stimulation of the metastatic tissue by TSH and/or hCG was responsible for this transient increase.
Major comments 5: As the authors know, Tg is not accurate in the presence of TgAb. Are all cases in this series negative for TgAb?
R5: According to reviewer's comment, the issue was highlighted in the legends to Figures 1 and 2. Moreover, we reported the sentence also in the new manuscript version (page 4, lines 161-162).

Reviewer 2 Report
Dear Authors,
This article focuses on an important topic. Since there is little data in the literature and considering the authors' experience in this field, this study is essential in identifying the guidelines that will underlie the best experimental and clinical practices. The authors provided adequate details on methodology, evaluation, findings, and investigations. The particularities and novelty of the article are very well underlined in the results and conclusions sections. Given the bibliography, it is clear that the authors made a complete review of the literature beforehand. Overall, this manuscript is well written and documented, representing an important moment in deciphering this pregnancy-related pathology. Indeed, the small number of cases could represent an impediment, but the cases were well justified.
However, some suggestions could improve the quality of the article:
- Effect of radioactive iodine on gonadal function.
- Some data related to the epidemiology of DTC in pregnancy should be entered
- Has next-generation DNA sequencing been performed, including common mutations in thyroid cancer?
- The lack of fetal outcome represents a minus of this article, which focuses only on the maternal outcome.
- What was the mode of delivery at the level of the studied group?
Kind regards
Author Response
Dear Authors,
This article focuses on an important topic. Since there is little data in the literature and considering the authors' experience in this field, this study is essential in identifying the guidelines that will underlie the best experimental and clinical practices. The authors provided adequate details on methodology, evaluation, findings, and investigations. The particularities and novelty of the article are very well underlined in the results and conclusions sections. Given the bibliography, it is clear that the authors made a complete review of the literature beforehand. Overall, this manuscript is well written and documented, representing an important moment in deciphering this pregnancy-related pathology. Indeed, the small number of cases could represent an impediment, but the cases were well justified.
However, some suggestions could improve the quality of the article:
Comment 1: Effect of radioactive iodine on gonadal function.
R1: We thank the reviewer for the comment. According to known data, most of the studies indicate that RAI therapy for DTC is not associated with a long-term decrease in pregnancy rates (Piek MW et al., The Effect of Radioactive Iodine Therapy on Ovarian Function and Fertility in Female Thyroid Cancer Patients: A Systematic Review and Meta-Analysis. Thyroid. 2021; 31(4):658-668. doi: 10.1089/thy.2020.0356.). Consistently, no gonadal function alterations and/or reduced fertility was observed in our patients (this was added to the revised text, page 5, lines 168, 169).
Comment 2: Some data related to the epidemiology of DTC in pregnancy should be entered
R2: In accordance with reviewer's suggestion, we added in introduction a sentence concerning the incidence of thyroid carcinoma in pregnancy (page 2, lines 46-48).
Comment 3: Has next-generation DNA sequencing been performed, including common mutations in thyroid cancer?
R3: We thank the reviewer for the suggestion. We had the opportunity to perform the genetic characterization by a customized panel on the tumour tissues of 3 cases; 1 case had BRAFV600E, 1 case had a double BRAFV600E and TERTc-124C>T mutation, and 1 case was wild type for the alterations characterized. This has been added to the methods and results sections (see page 3, lines 112-116 and page 5, lines 165-167).
Comment 4: The lack of fetal outcome represents a minus of this article, which focuses only on the maternal outcome.
R4: We thank the reviewer for the comment. There were no fetal complications in series analysed (see page 5, lines 168-169).
Comment 5: What was the mode of delivery at the level of the studied group?
R5: We thank the reviewer for the comment. All patients included in the study had eutocic deliveries. We therefore added the information in Table 1 and 2 and at page 5, lines 168-169.

Round 2
Reviewer 1 Report
The authors have revised the manuscript and answered almost satisfactory.
I expect further development of research. Thank you.
Reviewer 2 Report
Dear authors,
I reviewed this paper with particular interest because I investigated thyroid pathology in pregnancy, and I think it is important to emphasize the attention to the management of these pregnancies, given that the number of patients is relatively small. Nevertheless, we still need to conduct investigations in this specific area and collect more data in the future.
Kindest regards